# Discovery of a selective and biologically active low-molecular weight antagonist of human interleukin-1β

Ulrich Hommel [1] ✉, Konstanze Hurth [1] ✉, Jean-Michel Rondeau[1], Anna Vulpetti [1], Daniela Ostermeier[1], Andreas Boettcher[1], Jacob Peter Brady[2], Michael Hediger[1], Sylvie Lehmann[1], Elke Koch[1], Anke Blechschmidt[1], Rina Yamamoto[1], Valentina Tundo Dottorello[1], Sandra Haenni-Holzinger[1], Christian Kaiser[1], Philipp Lehr[1], Andreas Lingel [1], Luca Mureddu[3], Christian Schleberger [1], Jutta Blank [1], Paul Ramage[1], Felix Freuler[1], Joerg Eder [1] & Frédéric Bornancin [1] ✉

Human interleukin-1β (hIL-1β) is a pro-inflammatory cytokine involved in many diseases. While hIL-1β directed antibodies have shown clinical benefit, an orally available low-molecular weight antagonist is still elusive, limiting the applications of hIL-1β-directed therapies. Here we describe the discovery of a low-molecular weight hIL-1β antagonist that blocks the interaction with the IL-1R1 receptor. Starting from a low affinity fragment-based screening hit **1**, structure-based optimization resulted in a compound (*S*)-**2** that binds and antagonizes hIL-1β with single-digit micromolar activity in biophysical, biochemical, and cellular assays. X-ray analysis reveals an allosteric mode of action that involves a hitherto unknown binding site in hIL-1β encompassing two loops involved in hIL-1R1/hIL-1β interactions. We show that residues of this binding site are part of a conformationally excited state of the mature cytokine. The compound antagonizes hIL-1β function in cells, including primary human fibroblasts, demonstrating the relevance of this discovery for future development of hIL-1β directed therapeutics.

IL-1β is a key cytokine at the interface of innate and adaptive immunity[1,2]. It is produced by several types of myeloid cells upon stimulation by pathogen-associated molecular patterns (PAMPs; e.g., LPS, endotoxins) and danger-associated molecular patterns (DAMPs; e.g., urea crystals). IL-1β is produced as a 269 amino acid precursor protein, which is cleaved by molecular platforms known as inflammasomes that assemble under pro-inflammatory conditions, leading to extracellular release of its C-terminal 153 amino acid-containing mature form[3,4]. The binding of mature IL-1β to its cognate receptor IL-1R1 on nearby cells and subsequent formation of a ternary signaling complex with IL-1RAcP triggers a pro-inflammatory response with the production of inflammatory mediators, such as IL-6[4]. Owing to its central role in inflammatory processes, human IL-1β has been linked to multiple inflammatory disorders, such as rheumatoid arthritis, gout, periodic fevers, and neuroinflammation[2]. More recently, hIL-1β has also been linked to the pathogenesis of atherosclerosis and the progression of cancer[5–9]. This was further evidenced by the CANTOS study in which canakinumab, an anti-hIL-1β antibody[10], indicated the potential for interfering with hIL-1β to reduce cardiovascular risk[11,12].

[1]Novartis Institutes for BioMedical Research, Novartis Campus, CH-4002 Basel, Switzerland. [2]Novartis Institutes for BioMedical Research, 250 Massachusetts Avenue, Cambridge, MA 02139, USA. [3]Leicester Institute of Structural and Chemical Biology, Department of Molecular and Cell Biology, University of Leicester, Leicester LE1 7RH, UK. ✉e-mail: uli.hommel@icloud.com; konstanze.hurth@novartis.com; frederic.bornancin@novartis.com

While an anti-hIL-1β antibody-based approach has demonstrated substantial benefit in patients, an orally available IL-1β antagonist could allow for targeting a broader spectrum of diseases. In particular, there is increasing evidence for the contribution of inflammasomes to neuroinflammation, which cannot be studied well with antibody tools due to their limited passage through the blood–brain barrier[13]. Potent peptidic IL-1R1 antagonists comprising up to 21 amino acids were described, but evidence for their in vivo efficacy is lacking[14,15]. More recently, weak hIL-1β binders were identified by a crystallography-based fragment screening approach, but neither affinity nor functional activities have been reported for any of the fragments[16]. In summary, there is no low-molecular weight antagonist yet with the potential to recapitulate the benefits of hIL-1β antibodies.

Here, we describe the discovery of a low-molecular weight hIL-1β antagonist that blocks the binding of the cytokine to its cognate receptor IL-1R1 with an $IC_{50}$ in the single-digit μM range both in biochemical and cellular assays. Structural studies reveal that the antagonist binds to a hitherto unknown cryptic pocket on hIL-1β that involves residues at the interface with the third Ig-domain of the hIL-1β/IL-1R1 complex. In addition, the antagonist is functionally active at a cellular level, demonstrating the potential of targeting hIL-1β via low-molecular weight compounds for future therapeutic use.

## Results

### A fragment screen identifies the hIL-1β binder 1

To identify molecules that can bind to hIL-1β, we employed a fragment-based screening approach. A library of 3452 compounds containing $CF_3$, $CF_2$, or $CF$ moieties, known as LEF4000 (local environment of fluorine) library, was screened in mixtures by using standard $^{19}$F-NMR transverse relaxation measurements[17–19]. The degree of relaxation enhancement in the presence of protein was used as a criterion for hit nomination (Supplementary Table 1, Supplementary Fig. 1). A total of 16 fragments showed intensity changes by more than 30% in the presence of the target protein. These compounds were selected for further validation using standard protein-observed $^{1}$H–$^{13}$C- and $^{1}$H–$^{15}$N-HMQC NMR experiments. Only fragment 1 (Table 1, Supplementary Fig. 1b), a prototypic opener of the calcium-dependent potassium channel maxi-K[20], was validated as a binder by 2D NMR. Further characterization involved enantiomer separation of compound 1, yielding enantiomers (S)-1 and (R)-1, of which only enantiomer (S)-1 showed binding to the protein (Fig. 1a, b). The chemical shift perturbations indicated binding in the slow-exchange regime, where signal intensities rather than chemical shift positions of the affected resonances changed in a concentration-dependent manner. Fitting the concentration-dependent changes of line shapes and peak positions observed upon ligand binding to a simple 1:1 binding mechanism yielded a dissociation constant $K_D$ of 520 μM ($k_{on} = 1.2 \times 10^5$ M$^{-1}$ s$^{-1}$, $k_{off} = 63$ s$^{-1}$) for 1 (Supplementary Fig. 1c). The derived rate constants are thus one to two orders of magnitude slower than what one would expect for a fragment of low affinity, which indicated the existence of a slow conformational change governing the binding process.

### Mapping the binding site of 1 by NMR

For the identification of the ligand binding site, chemical shift changes of amide $^{1}$H- and $^{15}$N-resonances were mapped to the protein sequence. To this end, backbone resonances of hIL-1β under assay conditions were assigned based on those previously described at a different pH and temperature[21]. As shown in Fig. 1c, d, the largest chemical shift changes between free and ligand-bound hIL-1β were observed for residues lying in two opposing loops (loop β4–5, residues 47–55; loop β7–8, residues 87–96; throughout the text, amino acid numbers refer to the numbering of the proteolytically processed cytokine, in which Ala1 is equivalent to Ala117 of the unprocessed, native protein). The binding site of 1 is also in the vicinity of strand β5, whose hydrophobic residues (Ile56, Val58, Ala59, and Leu60) are similarly affected by

ligand binding in $^{1}$H–$^{13}$C–HMQC spectra (Fig. 1a). Notably, the resonances of many residues were broadened in the spectrum of the ligand-bound protein, presumably due to the intermediate/slow-exchange binding kinetics. When assessing the identified binding site in the context of the structure of the hIL-1β/IL-1R1 complex, we found that the residues affected upon binding of compound 1 cluster at the interface between domain 3 of the IL-1R1 receptor and hIL-1β (Fig. 1d). We thus embarked on the optimization of compound 1 to improve potency and probe the functional relevance of its binding site for inhibiting the hIL-1β/IL-1R1 interaction.

### Optimization of 1 into the functional hIL-1β antagonist (S)-2

The binding affinities of derivatives of 1 were initially assessed semi-quantitatively in protein-based $^{1}$H–$^{13}$C–HMQC NMR experiments, where the change in signal intensity of resonances was used as a measure for complex formation. To increase throughput, reduce protein consumption, and gain a quantitative read-out, we also developed a $^{19}$F-reporter-based displacement assay[17]. Together, the $^{13}$C-NMR and $^{19}$F-reporter data proved to be vital early on to support the structure–activity relationship (SAR) investigation on the parent fragment (compounds 3–9 in Fig. 2; Table 1, Supplementary Fig. 2a–f and Supplementary Methods). The indolinone NH and phenolic OH cannot be substituted with a methyl group (compounds 4 and 5). The lipophilic substituent in the 6-position of the indolinone (ring A) and in the 5-position of the phenol (ring C) also interact with the protein as their replacement with hydrogen decreases protein binding (compounds 6 and 7, Table 1). In contrast and as discussed below, extension from the chiral center and the phenol moiety of 1 was tolerated and offered opportunities to gain binding affinity (Table 1). Once compounds reached double-digit μM binding affinity, we employed a FRET-based receptor displacement assay. For selected compounds, we complemented the data set by measuring their binding affinities using surface plasmon resonance (SPR) (Supplementary Fig. 3a–d).

Details of the optimization process are summarized in Table 1 and Fig. 2. The $CF_3$ moiety in position 6 of ring A could be replaced, e.g., by a $CH_3$ group (compound 8, Table 1: $^{19}$F NMR $K_i = 312$ μM) or its Cl matched pair (compound 9, Table 1, $^{19}$F NMR $K_i = 172$ μM) with slightly increased binding affinity. Steric hindrance deep in the binding pocket limited substitutions at this position of the indolinone ring (vide infra). The strategy of extending from the phenol moiety (ring C) of 1 by replacing the halogen of the chlorophenol moiety with polar substituted phenyls led to compounds 10 and 11 (Fig. 2, Table 1) with an improved binding affinity to the cytokine ($K_i = 75$ and 63 μM for compounds 10 and 11, respectively). Furthermore, these compounds demonstrated a functional effect in inhibiting the cytokine receptor interaction (TR-FRET $IC_{50} = 178$ and 126 μM for compounds 10 and 11, respectively). Another region, which attracted the interest to interfere with the complex formation, was growing toward the loop β4–5 of IL-1β from the quaternary center in the 3-position of the indolinone (moiety D in Fig. 2). The exploration of functionalized aromatic moieties able to engage in a hydrogen-bond with Glu50 while making lipophilic contacts with sidechains of Pro57 and Val47 led to a substantial affinity gain, e.g., the 3-pyrazolyl analog 12 (SPR $K_D = 138$ μM, Supplementary Fig. 3) over its 3-hydroxy matched pair 13 (SPR $K_D > 1000$ μM).

A combination of the above-described modifications resulted in compound (S)-2 that binds to hIL-1β with single-digit μM affinity (SPR $K_D = 1.1$ μM; Supplementary Fig. 3e, f) and inhibits the interaction with its receptor IL-1R1 in the FRET-based assay at a similar potency (FRET $IC_{50} = 4.0$ μM, Fig. 3a); of note, none of the synthesized analogs of compound (S)-1 inhibited the IL-1R1/IL-1α interaction (Table 1). As observed for the initial hit, binding affinity to hIL-1β resided in the (S)-enantiomer of 2 only, while its (R)-enantiomer was devoid of functional activity in the TR-FRET assay and had no measurable affinity in biophysical assays. In addition, we studied the binding kinetics of (S)-2 by

**Table 1 | Biochemical, biophysical, and cellular activity data of derivatives of 1**

| Compound Id | Chemical structure | I/Io | $^{19}$F-reporter (µM) | SPR (µM) | TR-FRET IL-1β (µM) | TR-FRET IL-1α (µM) | RGA HEK293 (µM) | IL-6 (µM) |
|---|---|---|---|---|---|---|---|---|
| **1** | | 0.35 ± 0.06 (N = 2) | n. a. | n.d. | >200 | >200 | n.d. | n.d. |
| (S)-**1** | | 0.15 ± 0.01 (N = 2) | n.a. | n.d. | >200 | >200 | n.d. | n.d. |
| (S)-**2** | | <0.1 | <10 | 1.1 ± 0.1 (N = 3) | 4.0 ± 1.1 (N = 5) | >200 | 5.3 ± 0.3 (SEM, N = 8) | 7.9 ± 2.0 (SEM, N = 11) |
| (R)-**2** | | 0.97 ± 0.06 (N = 2) | >1000 | > 500 | >200 | >200 | n.d. | >100 (N = 2) |
| **3** | | 0.19 ± 0.03 (N = 2) | 151 ± 78 (N = 2) | 172 ± 2 (N = 2) | >400 | n.d. | n.d. | n.d. |
| **4** | | 0.98 ± 0.06 (N = 2) | >1000 | n.d. | >500 | n.d. | n.d. | n.d. |
| **5** | | 0.99 ± 0.06 (N = 2) | >1000 | n.d. | >530 | n.d. | n.d. | n.d. |
| **6** | | 0.59 ± 0.03 (N = 2) | >1000 | n.d. | >500 | n.d. | n.d. | n.d. |
| **7** | | 0.45 ± 0.05 (N = 2) | >1000 | n.d. | >500 | n.d. | n.d. | n.d. |
| **8** | | 0.18 ± 0.004 (N = 2) | 312 ± 124 (N = 58) | 261 ± 89 (N = 3) | >200 | >200 | n.d. | n.d. |
| **9** | | 0.3 (N = 1) | 172 ± 35 (N = 2) | n.d. | n.d. | n.d. | n.d. | n.d. |
| **10** | | 0.1 (N = 1) | 75 ± 40 (N = 2) | 44 ± 17 (N = 5) | 178 ± 38 (N = 3) | >200 | n.d. | 117 (N = 1) |
| **11** | | <0.1 (N = 1) | 63 ± 41 (N = 2) | 32 ± 5 (N = 5) | 126 ± 17 (N = 7) | >200 | n.d. | 112 ± 1 (N = 2) |

**Table 1 (continued) | Biochemical, biophysical, and cellular activity data of derivatives of 1**

| Compound Id | Chemical structure | I/Io | ¹⁹F-reporter (µM) | SPR (µM) | TR-FRET IL-1β (µM) | TR-FRET IL-1α (µM) | RGA HEK293 (µM) | IL-6 (µM) |
|---|---|---|---|---|---|---|---|---|
| 12 | | <0.1 (N = 1) | 90 ± 41 (N = 2) | 138 ± 11 (N = 2) | >200 | >200 | n.d. | n.d. |
| 13 | | 0.4 (N = 1) | n.d. | >1000 | n.d. | n.d. | n.d. | n.d. |

n.a. not applicable, n.d. not determined. I/Io is the ratio of the peak intensities of Leu60 C$^{δ1}$H$_3$ in ¹H-¹³C-HMQC spectra in the presence and absence of ligand, respectively. Compounds without stereodescriptor are racemates. Measurements were taken from distinct samples. Source data are provided as a Source Data file.

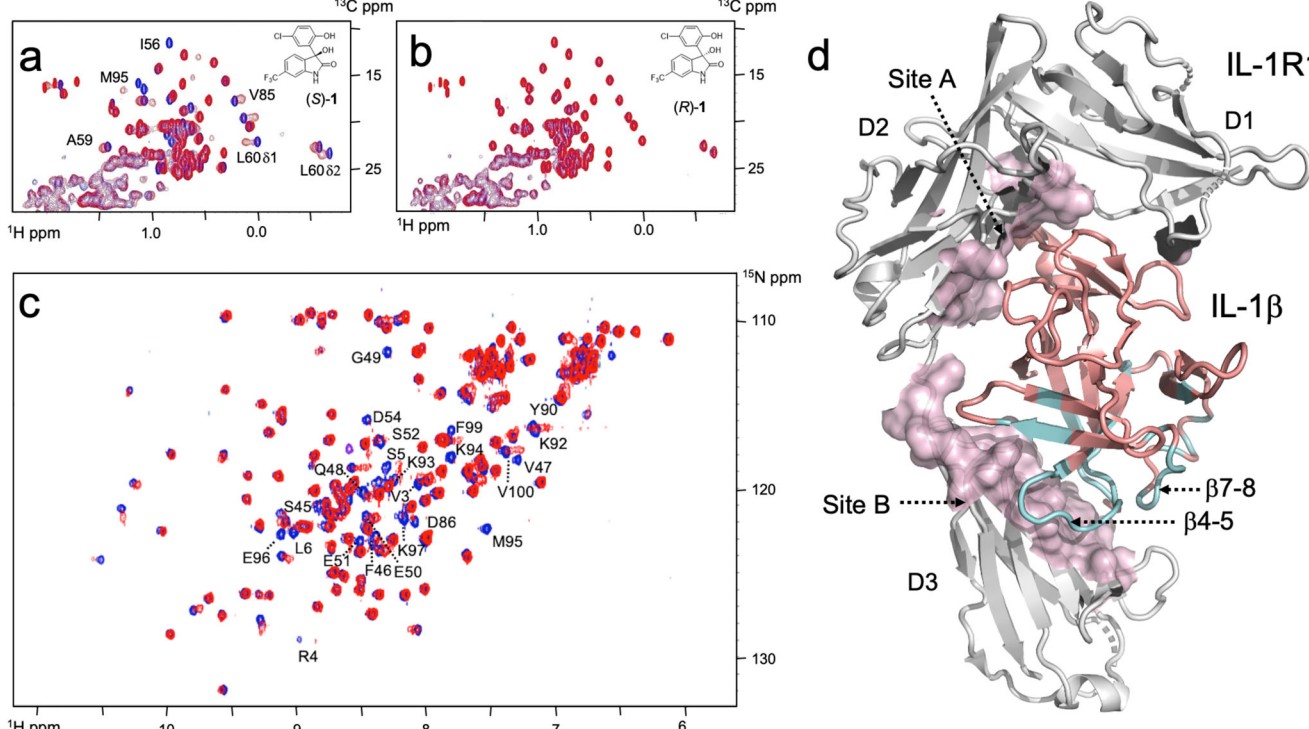

**Fig. 1 | Binding of fragment (1) to hIL-1β and mapping of its binding site.**
**a**, **b** Superposition of ¹H-¹³C-HMQC spectra for hIL-1β in the absence (blue) and presence of (S)-**1** and (R)-**1** (red). Residues that experience strong chemical shift changes upon the addition of (S)-**1** are labeled. **c** Superposition of ¹H-¹⁵N-HMQC spectra of hIL-1β in the absence (blue) and presence (red) of **1**. Some residues that experience strong chemical shift differences or residues with intensities below the detection limit due to chemical exchange phenomena are labeled. **d** Residues that are affected upon binding of **1** to hIL-1β in ¹H-¹⁵N HMQC spectra are mapped in cyan onto the X-ray structure of hIL-1β (PDB: 4DEP[31]). The affected residues cluster at the β-strands β1 and β5 and loops β4-5 and β7-8. Most of these residues are in close vicinity to domain 3 of the IL-1β/IL-1R1 interface, which represents site B of the interaction between IL-1β and its receptor.

SPR (Supplementary Fig. 3) and found that it is governed by a slow $k_{on}$ rate of $1.2 \times 10^5$ M$^{-1}$ s$^{-1}$ ($k_{off} = 0.12$ s$^{-1}$), a value we independently obtained also for the parent fragment (S)-**1** by NMR (Supplementary Fig. 1c). This indicated that the chemical derivatization of the parent fragment was effective in optimizing the interactions of the ligand with the protein in the bound state, while it did not have an influence on a ligand-independent process, preceding compound binding.

**Compound (S)-2 blocks IL-1R-mediated signaling in cells**
We next tested whether occupancy of the binding site described above could modulate hIL-1β-induced cellular activity, which relies on the heterodimeric receptor IL-1R1/IL-1RAcP. We first used an IL-6 release assay in human primary dermal fibroblasts. These cells express high levels of IL-1R1 and can release IL-6, which provides a robust readout, fully blocked by the anti-IL-1β antibody canakinumab (Fig. 3b).

Compound (S)-**2** could also block IL-6 release with an IC$_{50}$ in the single digit micromolar range, thus close to that obtained in the FRET-based hIL-1β/IL-1R1 interaction assay, which is in keeping with an extracellular mode of action (Fig. 3b, Table 1). As shown in Fig. 3c, both hIL-1α and hIL-1β can stimulate IL-6 release in these cells. The IL-1R1 antagonist anakinra could block the release of IL-6 triggered by concentration ranges of either hIL-1α or hIL-1β, whereas canakinumab selectively inhibited IL-1β driven activity, consistent with the respective mechanism of action of these two pharmacological principles[10,22,23]. Importantly, the compound (S)-**2** had no effect on hIL-1α driven IL-6 release, while it could inhibit the response to hIL-1β (Fig. 3c). To obtain further evidence of cellular activity, we used a reporter gene assay in HEK293 cells that reads out for IL-1 signaling. Compound (S)-**2** efficiently inhibited the reporter activity induced by hIL-1β but was ineffective when hIL-1α was used as a trigger. Canakinumab and anakinra, used as

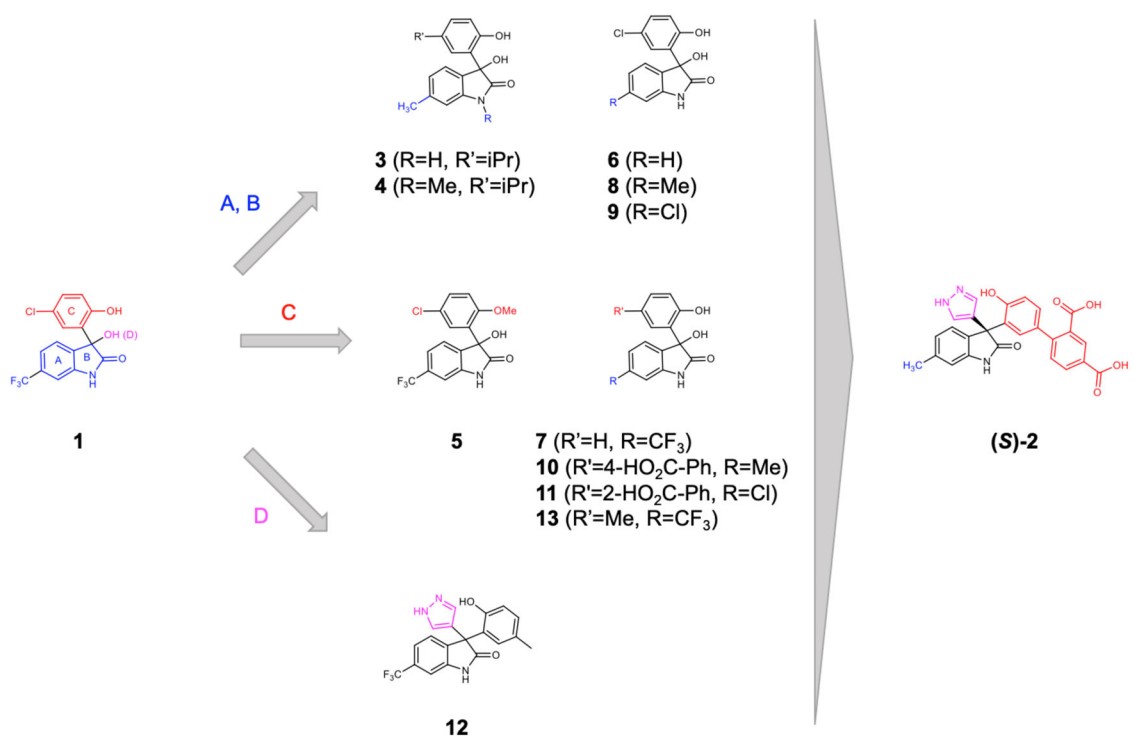

**Fig. 2 | Chemical optimization strategy.** Modifications in the indolinone and phenol moiety (rings A and C) and extensions from ring C and position D to probe interactions in the binding site and optimize hit **1** into the functionally active compound (*S*)-**2**.

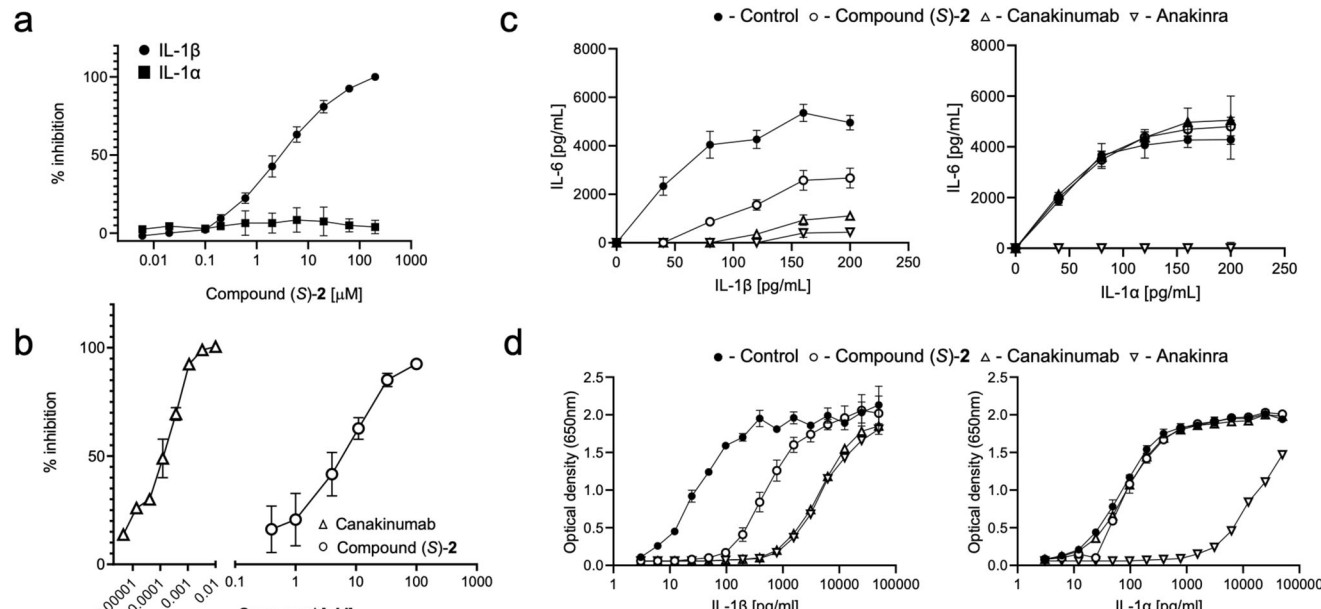

**Fig. 3 | Compound (*S*)-2 is a specific antagonist of the hIL-1β /IL-1R1 interaction. a** TR-FRET assay using recombinant hIL-1α (filled squares) or hIL-1β (filled circles) in the absence (control, 100%) or presence of graded concentrations of compound (*S*)-**2** (IL-1α, *N* = 2; IL-1β, *N* = 5). **b** IL-6 release in fibroblasts stimulated with 30 pg/ml IL-1β for 6 h in the presence of a concentration range of canakinumab (open triangles) or compound (*S*)-**2** (open circles); data are presented as % of maximal inhibition of IL-6 release (*N* = 3). **c** IL-6 release by human fibroblasts stimulated with graded concentrations of hIL-1β (left panel) or hIL-1α (right panel), in the absence (filled circles, control) or presence of 50 µM compound (*S*)-**2** (empty circles), 3 nM canakinumab (upward triangles), or 3 nM anakinra (downward triangle) (*N* = 5). **d** Reporter gene assay in HEK293 cells stimulated and treated as described in (**c**); hIL-1β (left panel), hIL-1α (right panel) (*N* = 3). Representative data are shown as mean ± SD; measurements were taken from distinct samples; full data are recapitulated in Table 1. Source data are provided as a Source Data file.

controls again here, responded as expected (Fig. 3d). Collectively, these data demonstrated that the structural changes in hIL-1β induced by the antagonist precluded interaction with the heterodimeric receptor on the surface of native cells. The data also indicated that the

assay flowchart, which relies on a binding assay followed by a biochemical assay devoid of the co-receptor necessary for hIL-1β-driven signaling, was able to identify compounds with a functionally relevant mechanism of action.

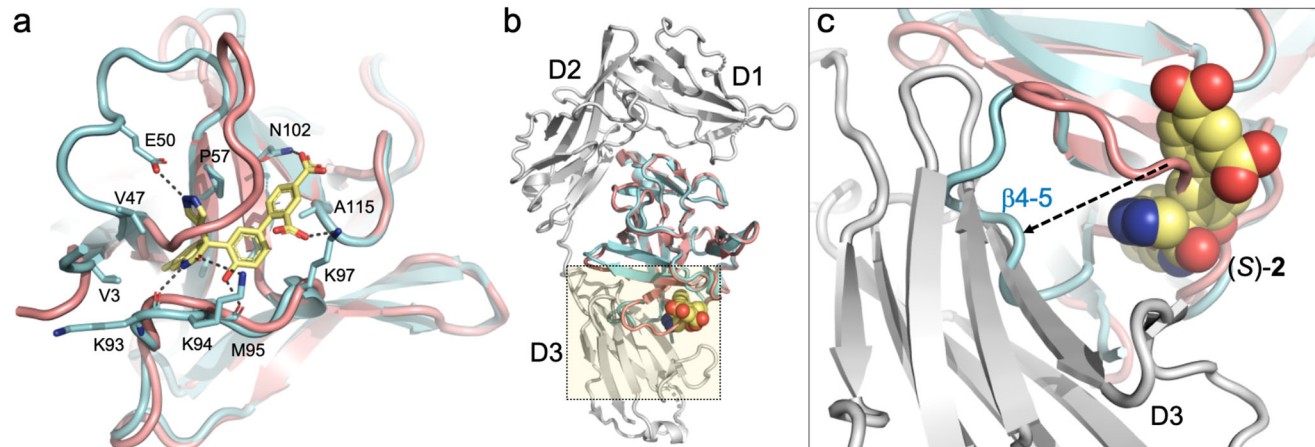

**Fig. 4 | Crystal structure of the hIL-1β/(S)-2 complex. a** Superposition of the hIL-1β/(S)-2 complex (cyan) with hIL-1β as observed in the ternary signaling complex with IL-1R1 and IL-1RAcP (salmon; PDB: 4DEP;[31] the receptor chains are not shown). The antagonist binds to a cryptic pocket formed by residues of the N-terminus, loop β4–5, and loop β7–8. H-bonded interactions between (S)-2 and IL-1β are shown with dashed lines. **b** Same overlay with IL-1R1 shown as a gray ribbon. **c** Close-up view of the region highlighted by the dotted square in (**b**). The orientation was changed to better show the displacement of loop β4–5 (residues 47–55) by up to 11 Å upon compound binding. The antagonist-bound conformation (cyan) is incompatible with proper engagement of the cytokine with domain 3 of IL-1R1.

## X-ray analysis reveals a cryptic binding site

The crystal structure of the hIL-1β/(S)-2 complex (Supplementary Table 2 and Supplementary Fig. 4) shows the binding of the ligand to a cryptic pocket formed upon displacement of loop β4–5 by up to 11 Å from its position in mature hIL-1β (Fig. 4a). The site is in good agreement with the binding site mapping of (S)-2 by NMR obtained in solution, where residues of loop β4–5, loop β7–8, and β-strand β5 are most strongly affected in the $^1H–^{15}N$–HMQC spectra (Supplementary Fig. 5). The X-ray structure shows a short hydrogen bond formed between the main chain carbonyl oxygens of Lys93 and Met95 and the phenol hydroxyl (2.5 Å) and the indolinone lactam nitrogen (2.8 Å) of the ligand, respectively. These two polar interactions anchor the ligand in the cryptic pocket and are thus critical interactions between the ligand and the cytokine, as shown by the lack of binding of the methylated analogs **4** and **5** (Table 1). The indolinone moiety (ring A, ring B) binds deep into the hydrophobic core of the cryptic pocket facing the side chains of Val3, Val47, Pro57, Met95, and Val100. The 6-methyl substituent is in van der Waals contact with Ser45 Cβ. Val100 makes extensive interactions with ring C (Fig. 4a). Ring C is also in van der Waals contact with the alkyl part of the Lys97 side-chain. The pyrazole group engages in a hydrogen-bonded interaction with the side-chain of Glu50, which is part of the loop β4–5. The para benzoic acid group is H-bonded to Asn102 $N^{δ2}$ and makes water-mediated interactions with Lys55 $N^ζ$, Lys97 $N^ζ$, and the main chain carbonyl oxygen of Ala115. The ortho benzoic acid group is H-bonded to Lys97 $N^ζ$ and forms a long ionic interaction with Lys94 in one of the two complexes making up the asymmetric unit of the crystal.

Notably, the structure of the hIL-1b/(S)-2 complex explains the high stereoselectivity of the original fragment **1**. Important interactions of compound (S)-2 are made by the C-ring, including the short H-bond (2.5 Å) mentioned above between the phenol hydroxyl and the main-chain carbonyl of Met-95. Only the (S)-enantiomer of the fragment, but not its (R)-enantiomer, can form this hydrogen bond together with hydrophobic interactions between the C-ring, Val100, and Lys97. Conversely, the para-chloro-phenyl moiety of the (R)-enantiomer would sterically clash with loop β4–5.

The analysis of the hIL-1β/(S)-2 structure allowed for an assessment of the mode of inhibition of hIL-1β. Two major interfaces are known to contribute to the binding of hIL-1β to its receptor IL-1R1. Site A encompasses residues 11, 13–15, 20–22, 27, 29–36, 38, 126–131, 147, and 149 of the cytokine, which are in close contact with IL-1R1 domain-1 and domain-2, while residues 4, 6, 46, 48, 51, 53–54, 56, 92–94, 103,

105–106, 108–109, 150, and 152 are primarily in contact with domain-3 of the receptor[24–26]. As shown in Fig. 4b, the binding of (S)-2 to hIL-1β did not perturb the conformation of residues at site A. In contrast, in the ligand-bound state, loop β4–5 adopts a conformation that is incompatible with the receptor-bound state of the cytokine at site B, which thus forms the structural basis for the antagonistic activity of (S)-2 and its analogs.

## A conformational equilibrium influences the binding of (S)-2 to hIL-1β

The discovery of a cryptic pocket, amenable to binding of low-molecular weight compounds, prompted us to consider whether this pocket is induced upon interaction with the compound or whether it already existed as a sparsely populated, so-called conformationally excited state[27,28], in solution. Earlier studies had indicated the existence of an equilibrium between a major and a minor form of hIL-1β in solution ($K$ = [major hIL-1β]/[minor hIL-1β] = 15; in sodium acetate buffer pH 5.4, 309 K)[21,29]. Since our studies were performed under slightly different conditions (PBS pH 7.4, 296 K), we conducted chemical exchange saturation transfer (CEST) NMR experiments to confirm the existence of this equilibrium and to further delineate the residues involved in the associated conformational exchange process[30] (Supplementary Fig. 6a, b). By mapping the residues involved in the formation of the minor form onto the X-ray structure of hIL-1β, it became apparent that these cluster at the N-terminus, in loop β4–5 and loop β7–8 of the protein (Fig. 5a, Supplementary Table 3), as previously seen by Clore et al.[29]. A quantification of the amount of the minor form was afforded by the CEST experiments, using data acquired at two different B1 fields ([minor hIL-1β] = 8% at 309 K; $k_{ex} = k_{+1} + k_{-1} = 22\,s^{-1}$; Supplementary Table 4). Intriguingly, the minor form involves residues Val47, Gln48, Lys92, and Lys94, which are all critically involved in the formation of the binding site of compound (S)-2. We thus considered the minor form to be potentially involved in compound binding and studied this hypothesis in more detail.

We first conducted the same CEST experiments on the hIL-β/(S)-2 complex to see if the compound's presence would influence the existence of the minor form. As shown in Fig. 5b (and Supplementary Fig. 7), in the presence of the compound, all residues that adopted two slowly interconverting conformational states without the compound were now stabilized in one state. The presence of the minor form and ligand binding are thus interdependent, which raised the possibility that this second conformational state of hIL-1β harbors the binding site

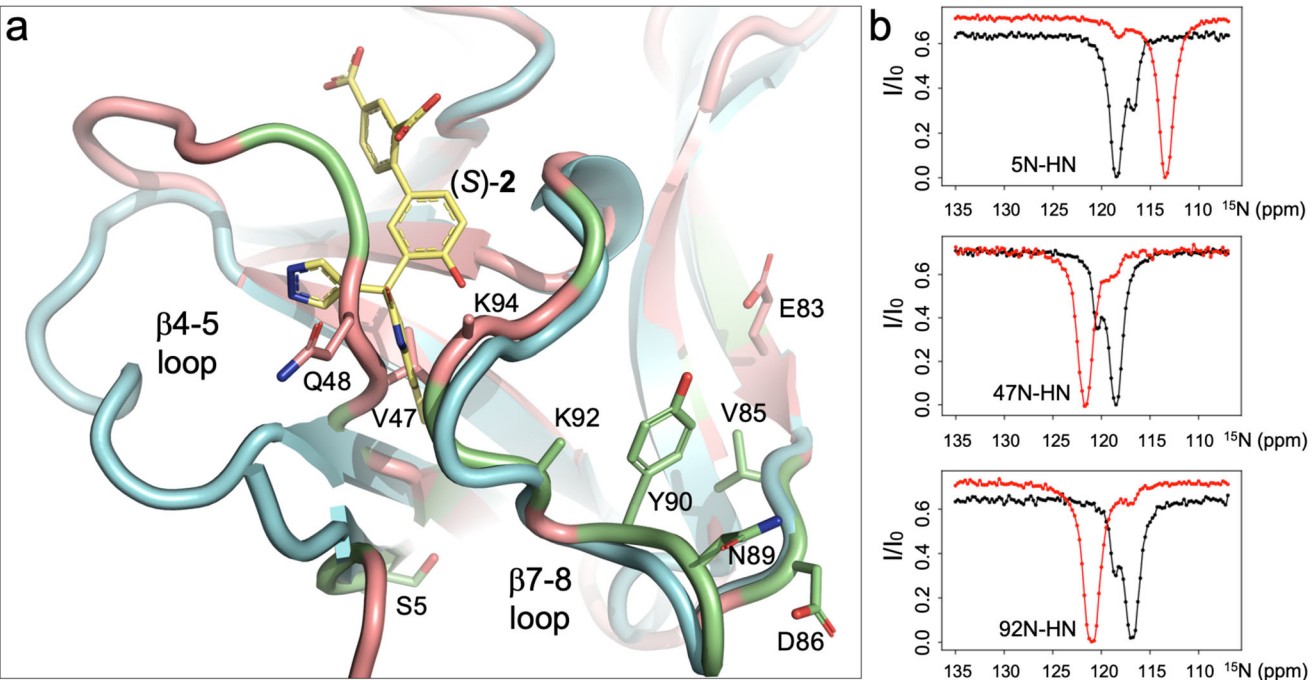

**Fig. 5 | Conformational heterogeneity of hIL-1β maps to the binding site of (S)-2. a** Superposition of the hIL-1β/(S)-2 complex (cyan) with hIL-1β as observed in the ternary complex with IL-1R1 and IL-1RAcP (salmon; PDB: 4DEP[31]). Residues previously[29] shown to be involved in a conformational exchange process are shown in lime green. Residues identified by CEST experiments in this study are shown in stick representation and labeled by their residue name. Note that due to high B-factors, the side-chain atoms of Lys92 and Lys94 were omitted beyond the $C^β$ atoms by the authors of PDB: 4DEP[31]. **b** [15]N-CEST profiles for selected residues of hIL-1β are

shown. In the absence of ligand (black), the traces show two dips representing a major form of the protein in slow conformational exchange with a minor form. In the presence of the IL-1β antagonist (S)-2, peaks move to the chemical shift of the ligand-bound species (red), and the second species disappears. Note that due to only 95% saturation under our experimental conditions ($K_D = 1.1$ μM, $P_0 = 0.8$ mM, $L_0 = 1.5$ mM), some residual intensity at the chemical shift of the unbound species is still visible.

for our compounds. We therefore asked the question of whether one could shift the equilibrium between the major and the minor form and hypothesized that amino acid substitutions in the cryptic pocket could destabilize the major in favor of the minor form and thereby facilitate access of ligands to the binding site. Inspection of the hIL-1β/(S)-2 complex indicated Val47 to be a critical residue in this sense: (a) it is centrally located in the cryptic pocket of unliganded hIL-1β; (b) it is shifted by 4.3 Å upon displacement by (S)-2 (Figs. 4a and 5a); (c) it is part of the conformationally excited state of the protein (Fig. 5b, d) it adds only minimally to (S)-2 binding (two long van der Waals contacts between Val 47 $C^{γ2}$ and the A and D rings of (S)-2 (3.8 Å and 3.6 Å, respectively). We thus generated the hIL-1β variant hIL-1β$^{V47A}$ and studied its conformational properties and compound binding.

To prove the structural preservation of this variant, we measured its affinity to hIL-1R1 and found it to be similar to that of hIL-1β (Supplementary Fig. 8). Furthermore, we established assignments for most backbone resonances of hIL-1β$^{V47A}$ as a basis to analyze its conformational properties by CEST experiments, as outlined above for wild-type hIL-1β. The chemical shift data further confirmed the structural integrity of the protein as only residues in the direct vicinity of the introduced change obeyed differences in their backbone resonances relative to the values in hIL-1β (Fig. 6a, Supplementary Fig. 9). The CEST experiments showed that several residues, which adopted a second conformation in wild-type hIL-1β, still do so in hIL-1β$^{V47A}$ (Supplementary Fig. 10, Ala47, Gln48, Asn89, Tyr90). However, the quantitative analysis revealed a ~3-fold shift of the equilibrium toward the minor form (hIL-1β$^{V47A}$: $K = 3.4$ hIL-1β: $K = 11.3$; Supplementary Fig. 11, Supplementary Table 3). Of note, due to the increased amount of the second species, we could now discern additional peaks corresponding to the minor form of the protein and putatively assigned their origin (Supplementary Fig. 12).

Together with the increase in the amount of the minor form, we observed a 5-fold reduction in the amount of hIL-1β$^{V47A}$ compared to hIL-1β for obtaining a >80 % reduction of the [19]F-signal of (S)-1 (Fig. 6b) in [19]F-transverse relaxation experiments. The amino acid substitution in the cryptic pocket thus not only shifted the equilibrium of hIL-1β toward its minor form, but it also generated the hIL-1β form capable of binding our ligands. Furthermore, while hIL-1β$^{V47A}$ was still capable of binding fragment (S)-1 with an affinity similar to that observed for hIL-1β, the binding process is now governed by moderately fast exchange kinetics, with resonances of the protein moving in ligand concentration-dependent manner (Fig. 6c, hIL-1β$^{V47A}$: $K_D = 263$ μM, $k_{on} = 0.81 \times 10^6$ M$^{-1}$ s$^{-1}$, $k_{off} = 215$ s$^{-1}$; Supplementary Fig. 13, hIL-1β: $K_D = 287$ μM, $k_{on} = 0.27 \times 10^6$ M$^{-1}$ s$^{-1}$, $k_{off} = 77$ s$^{-1}$). In line with this observation, both the on-rate and off-rate are increased, indicating the facilitated access of the ligand to the binding pocket and a potential loss of vdW contacts between Val47 $C^{γ2}$ and the A ring of (S)-1. Together, these data suggest that a conformational exchange process involving Val47 accompanies compound binding, as confirmed by X-ray analysis, and that this process involves the minor state of hIL-1β. The data thus provide a mechanistic link between the cryptic pocket and the minor state of hIL-1β.

## Discussion

hIL-1β belongs to a family of 11 cytokines that all share the same β-trefoil-fold structure despite divergent amino-acid sequences (IL-1α, IL-1Ra, IL-18, IL-33, IL-36α, IL-36β, IL-36γ, IL-36Ra, IL-37, IL-38). Except for IL-1α and the natural IL-1 receptor antagonist IL-1Ra, all other members bind to different receptors[31]. Structure determination of the IL-1R1 complexes with hIL-1β and the naturally occurring antagonist IL-1Ra, and the complex of hIL-1β with IL-1R1 and the associated receptor IL-1RAcP, reveal details about the molecular requirements for hIL-1β

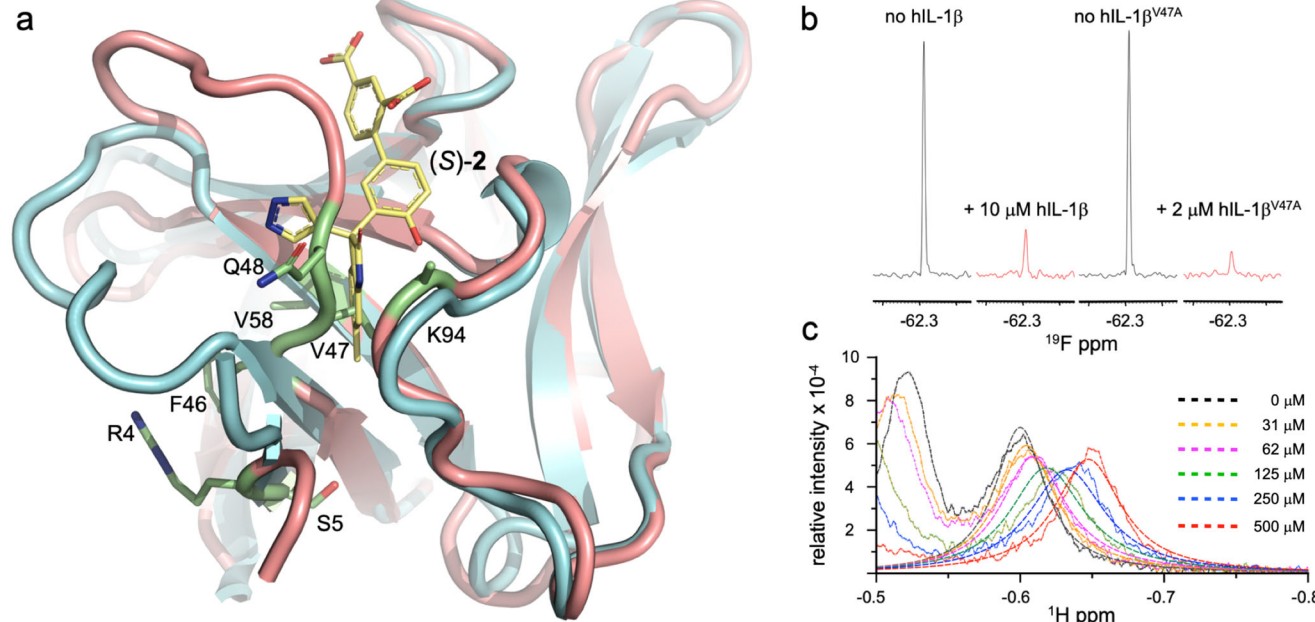

**Fig. 6 | Characterization of hIL-1β$^{V47A}$. a** Chemical shift differences of backbone $^{15}$N and $^{1}$H resonances (Supplementary Fig. 9) between wild-type and hIL-1β$^{V47A}$ are mapped in lime green onto the X-ray structure of hIL-1β (salmon; 4DEP[31]). The isopropyl side-chain of Val47 sits deep in a hydrophobic pocket utilized by ring A of (*S*)-**2** in the crystal structure of its complex with hIL-1β (this work; cyan).

**b** $^{19}$F-transverse relaxation experiment ($\tau$ = 240 ms) with compound (*S*)-**1** (40 μM) in the absence and presence of hIL-1β and hIL-1β$^{V47A}$. Signals were scaled to an internal reference. **c** Binding of (*S*)-**1** to hIL-1β$^{V47A}$. The protein was titrated with increasing amounts of the compound, and lineshapes were globally fitted to a two-site exchange model ($K_D$ = 263 ± 40 μM, $k_{off}$ = 215 ± 47 s$^{-1}$; $N$ = 2)[56].

driven signaling[24,25,31]. From these structures, it has become apparent that binding of hIL-1β to IL-1R1 requires engagement of the cytokine at two major binding sites (site A and B). Interestingly, the natural antagonist IL-1Ra has a markedly shortened loop β4–5 that does not allow it to engage with the receptor at site B while having unchanged interactions at site A. Hence, proper engagement of loop β4–5 with the receptor seems to be a requirement for productive hIL-1β signaling. The hIL-1β antagonists described here take advantage of this mechanism by impacting the conformation of the critical loop β4–5 of hIL-1β at the level of site B of the cytokine-receptor interface, thereby inhibiting proper interaction between the antagonist-bound cytokine and its receptor.

The closest structural and functional homolog of hIL-1β is IL-1α. We thus tested our antagonists for functional activity in a FRET-based IL-1α/IL-1R1 activity assay. In accordance with the sequence differences between hIL-1β and IL-1α around its binding site (Supplementary Fig. 14), we did not observe inhibition of IL-1α by (*S*)-**2** or any of its derivatives. The specificity for IL-1β was corroborated by cellular data monitoring IL-1α and IL-1β mediated signaling in HEK293 cells and in human dermal fibroblasts. The cryptic pocket targeted by our antagonists thus allows for cytokine specificity. The binding of these hIL-1β antagonists to even more distantly related members of the IL-1 family is, therefore, very unlikely.

Recently, a low-molecular weight antagonist (A-552) of IL-36γ has been described[32]. A-552 binds to human IL-36γ close to domain 3 of the respective cytokine/receptor complex at a site that is still ~15 Å apart from the site described here for the hIL-1β antagonist (*S*)-**2** (Supplementary Fig. 14). However, while the two antagonist/cytokine binding sites do not overlap, they suggest a similar mechanism of action of the respective antagonist. In analogy to the IL-1β/IL-1R1 structures, A-552 may disturb the proper engagement of the cytokine with its receptor by displacing domain D3 of the IL-36 receptor. Despite these similarities, both binding sites differ enough to obtain high selectivity. For instance, owing to the sequence differences between the IL-36 isoforms around this binding site (Supplementary Fig. 14b), A-552 has a high isoform specificity for IL-36γ and does not block IL-36α or

IL-36β[32]. Likewise, the sequence differences between IL-36γ and hIL-1β around this binding site preclude binding of A-552 to hIL-1β and A-552 does not inhibit hIL-1β[32]. Consistently, we found no inhibitory activity of the compound (*S*)-**2** against IL-36γ using a NanoBiT reporter system (Supplementary Fig. 15 and Supplementary Methods).

The existence of a conformationally excited state, which upon stabilization through low-molecular weight binders renders hIL-1β non-functional, allows for speculation on its biological relevance. Previously, residues 95-105 of pro-IL-1β have been shown to have lower H/D-exchange rates compared to the rest of the propeptide, suggesting protection of this sequence from solvent access in the precursor protein. In addition, this stretch contains a triple-E sequence (Glu99, Glu100, Glu101; pro-IL-1β numbering scheme) with electrostatic complementarity to the triple-K sequence (Lys92, Lys93, Lys94) in mature hIL-1β[33]. Interestingly, the triple-K sequence is located within loop β7–8, which defines one side of the binding pocket of our antagonists. It is thus conceivable that the hIL-1β precursor protein, which requires intracellular processing by Caspase-1 to yield the functionally active cytokine, may be kept in an autoinhibited state via internal stabilization of the open conformation. Such autoinhibitory mechanisms are well-documented for kinases, which require phosphorylation to adopt conformations accessible to substrates. An autoinhibitory mechanism might be advantageous for cytokines, too, in case of uncontrolled release from the cytoplasm. For the related IL-18 cytokine, an interaction of the propeptide with the mature protein could be evidenced[34], but efforts to demonstrate a similar interaction for hIL-1β have been unsuccessful so far due to the intrinsic instability of the precursor protein.

IL-1β has been considered 'undruggable'[35], and visual inspection of its surface failed to identify any cavity suitable for binding small molecules with high enough affinity. Only recently, an X-ray crystallography-based fragment screening identified several hot spots for small ligands with weak affinity binding to the surface of IL-1β, and their structures could be determined[16]. Of note, none of these binding sites overlap with the site described here. This can be rationalized by the crystallographic contacts made by loop β4–5 in the crystals used,

which precluded the conformational change to the open state required for binding fragments like compound **1**. In contrast, the [19]F-NMR-based fragment screening in solution does not restrict the conformational space of the protein and has identified compound **1**, which takes advantage of a conformational exchange process governing the access to a cryptic binding site.

The identification of 'hidden' or cryptic pockets, as druggable sites on targets of high medical interest, has become the subject of intense research efforts[36–38]. It has been speculated that higher B-factors in crystal structures could indicate sites where cryptic pockets may form[39]. For hIL-1β, such an analysis, based on the early X-ray analyses, would not indicate loop β4–5 and loop β7–8 to be part of a cryptic pocket. However, this type of analysis is intrinsically biased because residues taking part in crystal contacts have restrained flexibility as compared to their behavior in solution. NMR studies carried out in solution showed that residues Glu49 to Ile56, which are part of loop β4-5, possess reduced Lipari-Szabo order parameters as opposed to most other residues of the protein, suggesting a higher degree of internal mobility[29]. In addition, while hIL-1β was shown to adopt two slowly interconverting conformations in solution, the relevance of this equilibrium for drug discovery could not be anticipated. The discovery of fragment **1** and its optimization into the functional hIL-1β antagonist (S)-**2** thus demonstrate the importance of such equilibria and illustrate their potential for drug discovery.

The finding of a hitherto unknown binding site on hIL-1β adds to the growing number of cases where cryptic pockets play a role in the interaction between small molecule ligands and their drug targets[40–42]. Cryptic binding sites or alternate protein conformations have become an important aspect of modern drug discovery as they bear the potential to increase the druggable genome[43]. An in-depth analysis of 93 protein−ligand complexes in which cryptic pockets are involved has shown that 21 of them are characterized by large movements of secondary structural elements[36]. In many of those cases, high-affinity binders have been described, underscoring the potential of such pockets for drug discovery. A recent example of this concept was the characterization of the conformational landscape of c-abl kinase, a target for a diverse set of inhibitors with therapeutic application in leukemias[42]. This kinase appears to exist in two distinct conformationally excited states that differ substantially from the protein's ground state, each state being the target for a subclass of inhibitors. Future efforts in drug discovery should thus benefit from an in-depth analysis of protein states, which may allow for tapping into a larger chemical space for interfering with complex biological systems and thereby expanding the druggable genome[43].

## Methods

### Protein expression and purification

Throughout the text, we refer to hIL-1β as the proteolytically processed mature protein starting at residue Ala1, which corresponds to Ala117 of the unprocessed protein. Mature hIL-1β was expressed in *Escherichia coli* (pET17b, AmpR, BL21, DE3), induced overnight at 18 °C with 0.5 mM isopropyl β-D-1-thiogalactopyranoside (IPTG). The protein was purified following the method of Wingfield et al. [44]. *E. coli* cells were lysed in 8 volumes of 50 mM Tris pH 8.0, containing 5 mM each of benzamidine-HCl, EDTA, and DTT. After centrifugation (45 min at 16,000*g*), the supernatant, diluted 1:1 with 50 mM Tris pH 8.5, was applied to an XK26/10 column packed with Q-Sepharose HP (Cytiva), which had been equilibrated with the same buffer. The unbound was collected, ammonium sulfate (320 g/L) was added slowly, and the solution was stirred overnight at 4 °C. After centrifugation (16,000*g*; 15 min), the pellet was discarded, and to the supernatant, ammonium sulfate was added to 77% saturation (176 g/L). The suspension was stirred overnight at 4 °C and centrifuged (16,000*g*; 45 min); the resulting pellet was then carefully resuspended in 50 mM MES pH 5.7 (buffer B) and finally dialyzed against 3 × 80 volumes of the same

buffer. The solution was filtered (0.45 μm) and applied to an XK16/20 column of SP-Sepharose HP, which had been equilibrated with buffer B. The column was washed with 5 column volumes of buffer B and eluted with a gradient of 0–1 M NaCl in buffer B over 15 column volumes. hIL-1β was eluted as a sharp, dominant peak and further purified using an XK16/60 column of Superdex75 equilibrated with 20 mM Tris pH 8.0, containing 100 mM NaCl. The final concentration of the protein was 7.5 mg/mL. For uniform labeling of proteins, [13]C-D-glucose, and [15]N ammonium chloride were used as isotope sources.

hIL-1β[V47A]-His6: Mature hIL-1β[V47A] was expressed with a C-terminal His6 tag and induced overnight at 20 °C with 0.1 mM IPTG. *E. coli* cell pellets from 1 L culture were resuspended in 50 mM NaH2PO4, containing 300 mM NaCl and 20 mM imidazole (buffer C). EDTA-free protease inhibitor tablets were added (cOmplete™ Mini, Roche, 1 tablet/50 ml), and the suspension was lysed and centrifuged as above. The filtered (0.45 μM) supernatant was applied to a 5 ml His-Trap column (Cytiva), which had been equilibrated with 4 column volumes of buffer C. The column was subsequently washed with 10 column volumes of buffer C and then eluted with a 0–100% gradient of buffer D (buffer C containing 0.5 M imidazole) over 15 column volumes. The eluted protein was concentrated to 5 ml using an Amicon Ultra 15 centrifugal concentrator (cut-off 3 kDa) and applied to an XK16/60 column of Superdex 75, equilibrated with PBS, pH 7.4. The protein eluted as a single peak and was aliquoted (1 ml) and stored frozen at −80 °C.

Interleukin-1R1(21–337)-His6: Interleukin-1R1(1–337)-His6 was expressed transiently in HEK-ExpiGnti⁻ cells. After 5 days, the cell suspension containing the mature Interleukin-1R1(21–337)-His6 was centrifuged for 20 min at 4136*g* to remove cells, and the resultant supernatant was filtered through a 0.45 μm Sartobran capsule. Purification was carried out using a 5 ml HisTrap crude column (Cytiva) which had been equilibrated with 50 mM NaH2PO4 pH 8.0 containing 300 mM NaCl, 20 mM imidazole and 10% v/v glycerol (buffer E). The unbound proteins were washed out with 6 column volumes of buffer E, and the bound protein was eluted with a gradient of 0–100% buffer F (buffer C containing 300 mM imidazole) over 8 column volumes. Fractions were analyzed by SDS-PAGE, and the appropriate fractions were pooled. The pool was concentrated to 4 ml and loaded onto a Superdex75 XK16/60 column equilibrated with 50 mM Tris pH 8.0, 50 mM NaCl, and 10% glycerol. Fractions were pooled following analysis by SDS-PAGE and concentrated to ~5 ml, giving a final concentration of 3.6 mg/mL.

hIL-1α(113–271): hIL-1α(113–271) and hIL-1α(113–271)-avi (sequential numbering of pro-IL-1α, Uniprot entry P01583) were both expressed with an N-terminal hexa-histidine-SUMO tag. Both proteins were purified over 5 ml HisTrap Crude columns (Cytiva) as described above for hIL-1β[V47A]-His6. The pooled HisTrap eluates were incubated overnight at 4 °C with ULP-1 protease (1:60 mass ratio), concentrated to 4 ml, and finally purified over Superdex75 (XK16/60) in PBS buffer pH 7.4. Pooled fractions with a purity greater than 95% by RP-HPLC were aliquoted and frozen at −80 °C.

### Fragment screening

A library consisting of 3,452 fragments was screened in mixtures as described[19]. Experiments were recorded on a Bruker AVANCE III 500 MHz NMR spectrometer equipped with a cryocooled 5 mm CP-QCI-F probehead. The protein concentration was 4 μM in PBS buffer at pH 7.4, 296 K. The fragments were assembled in mixtures of 32 and 30 compounds for CF3- and CF-containing fragments, respectively. The final concentration of individual compounds was 25 and 40 μM for CF3− and CF-containing fragments, respectively. Fluorine CPMG relaxation experiments were recorded with 240 ms and 120 ms on the CF3− and CF-mixtures, respectively. Two mixtures contained CF-CF2-containing compounds at 30 compounds per mixture. They were treated in the same way as the CF-containing mixtures. NMR spectra

were analyzed for binders using the Screen module in AnalysisScreen 3.0.4 (ref. [45]). The hit selection criterion was I/Io < 0.7, with I and Io representing the intensities of the $^{19}F$ signal in the presence and absence of protein, respectively.

## NMR spectroscopy

$^1H–^{13}C$- and $^1H–^{15}N$-HMQC spectra were recorded on Bruker AVANCE 600 MHz and 800 MHz spectrometers equipped with TXI-cryocooled probeheads. In ligand binding experiments, protein concentrations were typically 50 μM in PBS, pH 7.4. Backbone assignments of hIL-1β in the ligand-bound state and unliganded state were obtained by a combination of a 3D triple resonance experiment HNCACB and a $^{15}N$-edited NOESY spectrum, which were processed using TOPSPIN 3.2 and analyzed using CcpNmr AnalysisAssign 3.1.0[46]. $^{15}N$-CEST experiments were run at weak B1 radiofrequency fields of 25 and 12.5 Hz on a Bruker AVANCE 800 MHz spectrometer with 113 increments and a total sweep width of 28 ppm in the pseudo-3D dimension. Peak intensities were extracted by fitting pseudo-Voigt lineshape models using peakipy (https://github.com/j-brady/peakipy). Fitted peak intensities were subsequently used to generate CEST profiles, and exchange parameters were fitted using ChemEx (https://github.com/gbouvignies/ChemEx).

## $^{19}F$ reporter assay

Relative compound affinities were monitored in a reporter-based displacement assay using (S)-1 as the probe. $^{19}F$-NMR spectra were recorded with a mixing time $\tau = 240$ ms. Protein and reporter concentrations were 10 and 40 μM, respectively. To account for changes in sample homogeneity, samples contained 1-(2,2,2-trifluoroacetyl) piperidine-4-carboxylic acid as an internal reference for scaling signal intensities. Three concentrations of the competitor were typically measured, and the related displacement of the probe was converted into $K_i$-values according to Dalvit and Vulpetti[17]. For the reporter, a $K_D$ of 250 μM was initially derived from simulating the concentration-dependent lineshapes of compound 1 in a titration experiment and used throughout the hit optimization process for consistency reasons. This value is in good agreement with the one obtained later from fitting lineshapes ($K_D = 287$ μM; Supplementary Fig. 13). An estimated $K_i$-value for a given compound was calculated from the average obtained for three different concentrations of competitor (50, 150, and 450 μM). This process was automated with FRPipe, an in-house developed Python script (https://github.com/Novartis/FRPipe) that allows for streamlined analysis of such experiments. In brief, it reads recorded spectra using nmrglue[47], determines relevant reference and reporter peak volumes, retrieves data about protein, reporter concentrations, and reporter $K_D$, and uses these data points to fit the normalized, relative reporter signal to the theoretical curve describing the bound reporter fraction to $K_i$ relation[48]. For visual inspection, an overview containing relevant $^{19}F$ peaks, fitting curves, and proton spectra was plotted.

## TR-FRET assay

For IC50 determination, recombinant His-tagged wt IL-1β or IL-1α was diluted to 2.5 nM in assay buffer (20 mM Tris pH 7.5; 150 mM NaCl; 1 mM EDTA; 0.5% Pluronic and 0.02% BSA) and incubated in Greiner high base, low volume, black 384w plates (Greiner; Cat. No. 784076) with different concentrations of compound. After 1 h incubation at room temperature, hIL-1R1 (1–332; Alexa 647 labeled; 50 nM final concentration) and anti-His-Europium W1024 (Perkin Elmer; Cat.No. AD0401, 1 nM final concentration) were added to a final volume of 10 μl per well. After 1 h of incubation at room temperature, time-resolved fluorescence was measured at $\lambda_{exc} = 320 \pm 20$ nm, $\lambda_{em} = 620 \pm 25$ nm/665 ± 25 nm using a Pherastar FSX plate reader. All experiments were run in triplicates. The data provided a typical saturation-binding curve, and the IC50 values were calculated by fitting the results to the Hill equation using Prism 7.0 and the in-house software Helios.

## Surface plasmon resonance

All SPR experiments were performed with an SPR-32 instrument (Bruker Daltonics SPR, Hamburg, Germany). SPR chips were from XanTec bioanalytics GmbH (Düsseldorf, Germany).

**Binding of compounds to hIL-1β.** C-terminally avi-tagged hIL-1β protein was immobilized onto a SAHC200M chip with a neutravidin-coated surface. Immobilization was performed with a protein injection of 0.02 mg/ml IL-1β diluted in PBS pH 7.5 (Gibco 70011-036, 10×), 0.05% Tween-20 for 5 min at 10 μl/min. The running buffer for binding studies of compounds was PBS pH 7.4, 0.05% Tween-20, and 2% DMSO. For the correction of small DMSO variations between samples, injections of 6 solvent samples ranging from 1% to 3% DMSO were run. Compound measurements were performed in MICK mode (multiple injection cycles kinetic) with a dose–response of 10 concentrations. Dissociation constants for compounds ($K_D$) were obtained by determining steady-state fit that followed the 1:1 Langmuir model.

**Binding of IL-1β to IL-1R1.** The receptor IL-1R1 was covalently immobilized to the chip surface of a CMD50L sensor by amine coupling. The coupling reagents (EDC/NHS and ethanolamine) were from Bruker. 1× PBS pH 7.5 (Gibco 70011-036, 10×), 0.05 % Tween-20 was used as running buffer. The chip surface was activated with a 3 min injection of EDC/NHS (200 mM 1-ethyl-3-(3-dimethylaminopropyl)-carbodiimide hydrochloride, 25 mM N-hydroxysuccinimide), followed by the protein injection of 0.02 mg/ml IL-1R1 diluted in 10 mM sodium acetate pH 5.0 for 3 min at 10 μl/min. The remaining active succinimides on the surface were deactivated by an injection of 1 M ethanolamine hydrochloride pH 8.5 over 3 min at 10 μl/min.

The injection of the protein dilution series was performed with 8 concentrations in SICK mode (single injection cycle kinetic) with a flow rate of 40 μl/min and 2 reference injections with buffer before and after the protein injection.

Data analysis was performed with the Sierra Analyzer 3.4.0 software from Bruker. Signal intensities were corrected for nonspecific binding to the surface by subtracting intensities from the reference surface from those of the hIL-1β surfaces (first reference subtraction) and by subtracting the buffer injection (second reference subtraction). $K_D$ values were carried out by determining a reporting point close to the end of the injection, used for steady-state fit, in addition to a kinetic fit delivering $k_{on}$- and $k_{off}$-values; both fitting models followed the 1:1 Langmuir equation.

## Crystallization

hIL-1β (28.5 μl at 24 mg/ml in PBS) was mixed with (S)-2 (1.5 μl of a 100 mM stock solution in 90% DMSO-D6, 10% D2O) and crystallized by the vapor diffusion in sitting drops technique in Innovadyne SD-2 96-well plates at 20 °C, by mixing 0.2 μl of protein stock with 0.2 μl of crystallization buffer (0.01 M sodium citrate, 33% PEG 6,000) and equilibrating against the same buffer. Crystals appeared after 4 days and grew to full size within 24 h.

## X-ray structure determination

One crystal was directly mounted in a cryo-loop and flash-cooled by immersion into liquid N2. X-ray data were collected at the Swiss Light Source, beamline X10SA, with an Eiger pixel detector, using 1.000036 Å X-ray radiation. In total, 2700 images of 0.10° oscillation each were recorded at a crystal-to-detector distance of 260 mm and processed with autoPROC 1.1.7 (20220203)[49]. Isotropic analysis using $CC^{1/2}$ statistics (>0.3) led to a resolution cut-off of 1.945 Å (2.250 Å with $I/\sigma(I) \geq 2.0$)[50,51]. The structure was determined by molecular

replacement with Phaser 2.8.3[52] using an *in house* structure of human IL-1β as a search model. The ligand was built into strong difference electron-density ($\sigma_A$-weighted mFo-DFc map computed by auto-BUSTER 2.11.8 (20220203)), and the structure was refined against data from classical isotropic treatment and analysis, by multiple cycles of electron-density inspection and model rebuilding in Coot 0.9.6 EL[53], followed by automated refinement with autoBUSTER 2.11.8 (20220203)[54]. Feature-enhanced maps and Polder omit maps were calculated with Phenix 1.20_4459[55].

### HEK293 reporter gene assay

We used HEK-Blue™ IL-1β cells that respond specifically to IL-1 (# hkb-il1b, InvivoGen). These cells can detect IL-1α and IL-1β, as these cytokines bind to the same receptor, IL-1R1. They express a *Seap* reporter gene under the control of the IFN-β minimal promoter fused to five NF-kappaB and five AP-1 binding sites. Thirty micro-liters (300,000 cells/ml) per well of a 384-well plate were plated overnight. Cells were then stimulated with graded concentrations of cytokines (hIL-1α or hIL-1β) in the absence or presence of cana-kinumab (Ilaris®, ILARI LYVI 150MG 22GLW.012, batch U003 0409)[22], anakinra (Kineret®, 100 mg/0.67 ml injectable solution, Sobi)[23], or compound (*S*)-**2** (compounds were pre-mixed for 1 h with the cytokine prior to addition to cells) and further incubated overnight before assessment of *Seap* activity.

### Human fibroblast IL-6 release assay

Normal human dermal fibroblasts (NHDF) from Clonetics™ (LONZA, Cat # CC-2509 (NHDF-Neo), Lot # 0000234242, used at passage ≤14) were seeded at a density of 5000 cells/well in 100 µL of culture medium using 96-well plates (flat bottom, TCT). Treatment with cytokines/compounds was performed as described above. Cells were incubated for another 6 h and were subsequently spun down (320*g*/3 min). IL-6 was measured in the supernatant by ELISA (BioLegend, # 430503) or HTRF (Cisbio, #62HIL06PEH).

### Reporting summary

Further information on research design is available in the Nature Portfolio Reporting Summary linked to this article.

## Data availability

The data that support this study are available from the corresponding authors upon request. Diffraction data and crystallographic coordinates of the complex between hIL-1β and compound (*S*)-**2** generated in this study have been deposited in the PDB database under accession code 8C3U. Previously published crystallographic structures used in this study are available from the PDB under accession codes 4DEP (human IL-1β complex with IL-1R1 and IL-1RAcP), 2I1B (unliganded human IL-1β), 2KKI (NMR structure of human IL-1α) and 6P9E (human IL-36γ in complex with A-552). The NMR chemical shift assignments generated in this study have been deposited in the BMRB database under accession codes 51919 (hIL-1β at 309 K), 51859 (hIL-1β at 296 K), 51938 (hIL-1β in complex with (*S*)-**2**) and 51950 (hIL-1β$^{V47A}$ at 296 K and 309 K). Source data are provided in this paper.

## Code availability

The in-house $^{19}$F NMR data analysis script FRPipe is provided as a separate Supplementary Software file; the source code is also available at https://github.com/Novartis/FRPipe.

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

## Acknowledgements

We would like to thank Cesar Fernandez, Adeline Unterreiner, Tanja Kobel, Eva Langhammer, Patrick Scheifele, Alessio Scalone, Thomas Lochmann, Corinne Marx, and Aleksandar Stojanovic for technical support, Klemens Kaupmann for generating the CHO *Il1r* and CHO *Il36r* reporter lines, and Wolfgang Jahnke, Claudio Dalvit, Paul C. Driscoll and Lewis E. Kay for critically reading the manuscript. L.M. acknowledges the funding of the CCPN project by the MRC (grants 554 MR/L000555/1 and MR/P00038X/1).

## Author contributions

F. F. cloned the constructs. P.R., S.H.-H., V.T.-D., C.K., and S.L. purified the proteins. S.L. and E.K. crystallized the proteins. J.-M.R. determined the crystal structure. A.V. designed the LEF4000 library. U.H. and A.L. conducted the fragment-based screening. U.H., D.O., A.Bl., J.P.B., and A.L. performed NMR experiments. D.O. performed SPR experiments. J.B., R.Y., and A.Bo. developed and performed biochemical assays. F.B. designed and supervised the cellular assays; A.V. and K.H. designed the compounds. K.H., M.H., and P.L. synthesized compounds. C.S. and L.M. provided IT, software, and data analysis support. U.H. conceived the study; A.V. conceived the site-directed mutagenesis study. U.H., K.H., F.B., A.V., A.Bo., J.E., and J.-M.R. defined the project strategy and wrote the paper with substantial contributions from all the other authors.

## Competing interests

All authors except L.M. are current or former employees and shareholders of Novartis. L.M. declares no competing interests.
