## [Peer Review File · Nature Communications]

REVIEWER COMMENTS

Reviewer #1 (Remarks to the Author):

- What are the noteworthy results?

Hommel et al. report the discovery of the first low molecular weight antagonist of human interleukin-1 β . Despite the existence of protein antagonists of IL-1 β signaling that are currently in clinical use (e.g., Anakinra), no small molecule antagonist (with its potential for oral administration) has been reported to date. The authors characterized binding of the antagonist to the cytokine, discuss possible molecular basis for the inhibition and assess its activity through cell-based assays.

- Will the work be of significance to the field and related fields? How does it compare to the established literature?

Probably not. While the authors state that it is the first report of a low molecular weight antagonist of the IL-1 β signaling. However, several mitigating factors exist: (1) most importantly, the novelty of these findings is substantially reduced by a publication from a few years ago (Reference 27 in this manuscript) reported a small molecule inhibitor of IL-36 γ signaling, which binds in a very similar site on the cytokine as does these authors' small molecule on IL-1 β – the larger IL-1 family of cytokines includes the IL-1 and IL-36 cytokine sub-families, which make similar signaling complexes; (2) a number of protein-based antagonists already exist and are in the clinical use; and (3) the modest IC50 (low micromolar) of the compound and lack of in vivo efficacy/toxicity data suggest a long road to the development of a compound that be of significance to the field or related fields.

- Does the work support the conclusions and claims, or is additional evidence needed?

No. Concerns, both major and minor, listed below:

Line 46: “conformationally excited” – do the authors mean conformationally dynamic? Flexible? Excited suggests that there is some force acting upon this part of the molecule.

Line 119: Conventionally, the loops in IL-1 family cytokines are not labeled with letters, but with the following nomenclature: e.g., β 4-5 loop (i.e., the loop between β strands 4 and 5).

Figure 2B: The orientation of the IL-1 β /IL-1RI complex shown is strange and has no bearing to the membrane. It is conventionally shown with the D3 domain of IL-1RI at the bottom (as if coming out of a membrane below it).

Lines 134-140: Is there really no data shown for all of these binding experiments? There appears to only be a summary of the data in Table 1. It is therefore impossible to judge whether the data is valid.

Table 1: not immediately clear what parameter is shown.

Lines 144-145: “The detailed description of the structure-based design effort and SAR will be presented elsewhere”. Is this a joke? Are we supposed to just skip over the middle half of the paper and perhaps one day (or never) read about it elsewhere?

Fig. 3 B and C: The authors test their inhibitor biological activity in the cell-based assays in which they compare it to the existing biologics. What is the rationale for selection of the concentrations of all the antagonist for the assay? It would make more sense to test the same range of concentrations of particular reagents against a constant concentration of the cytokines. This would allow them to establish and compare IC50 values of all the tested agents.

In Figure 3, a legend identifying the data points would be helpful.

Lines 213-225: The authors list all of the interactions between their small molecule and IL-1 β that they observe in their crystal structure of the complex. They ascribe weight to some (“Key hydrogen bonds...”). However, they have no data indicating which intermolecular interactions are energetically important. To do so, they would need to either make a series of site-directed mutants in IL-1 β or derivatives of their small molecule in which individual functional groups were removed, and then measure binding affinities. Without such an analysis, it cannot be said that they have defined the molecular basis of this intermolecular interaction.

Figure 5 title: “excited” is not an appropriate descriptor here. The authors data indicate that IL-1 β exists in an ensemble of conformations, of which there are predominantly two – a “major” and a “minor” conformation, distinguished largely by the positions of residues in the loop starting at Val47. This region of IL-1 β may be conformationally flexible or dynamic, but it is not excited.

Line: 505: from previous work it seems that number of residues involved in site A and site B is similar

- Are there any flaws in the data analysis, interpretation and conclusions? - Do these prohibit publication or require revision?

See remarks above.

- Is the methodology sound? Does the work meet the expected standards in your field?

See remarks above.

- Is there enough detail provided in the methods for the work to be reproduced?

Yes.

Reviewer #2 (Remarks to the Author): - A pdf version of this report with the missing symbols included has been attached to this mail. -

In this manuscript, Hommel et al. reported the structural and functional studies of a novel small molecule inhibitor against human IL1 β . Overall, the study is thorough with extensive biophysical and functional analyses. The result is very exciting because this is the first report to date a functional specific small molecule compound was developed targeting hIL1 β . The authors showed that the inhibitor allosterically binds to the cytokine in an isoform specific manner and blocked IL1 β signaling. The study provided new insights into structure based inhibitor design against human disease related to IL1 β pathway.

1. An insightful discussion on how the cytokine discriminate the stereoisomers of the inhibitor 1, and how (s)-2 discriminate beta from alpha IL-1 would be beneficial to the audiences.
2. In general, use x in scientific notations and use comma as thousands separator.
3. In general, use superscripts for specific sidechain atoms. i.e. Lys-55 N⁺. Use main chain carbonyl oxygen for C=O of a residue and amide for HN.
4. Either use 'side-chain' or 'sidechain', be consistent throughout.
5. Line 254, 43 Å.
6. All kinetic and binding values are missing standard deviations, such as K_i, K_d, IC₅₀ etc.
7. SPR titration data not shown.
8. Fig.2A, label Loops D and G with pointer or similar to make them better discernable.
Fig.2B, add loops D and G, binding site A, B that were mentioned in the main text.
9. Fig. 4A, indicate the 11 Å swing. Use a different color scheme for (s)-2 from the protein. Label ring C. M95 and other residues described in contacts are not visible in the figure.
10. Fig. 5. Label loop D
11. Table S1. Add wavelength. Keep only 1 digit after decimal point for cell contents. List beta angle only. Keep 1/100 Å precision for resolution. 1/1000 for R values. Why B201 has much lower B than A201? List Wilson B. Define R_{work}/R_{free}.
12. The Methods section seems very roughly written and needs careful proof reading, it contains numerous typos. Here only list some:
Line 563, italicize E.coli.
Line 564/570/597/599, β C
Line 574/584, β m

Line 585, 682, leave space between a number and its unit, i.e. 50 mM, 5 ml, 260 mm

Line 579, what was the concentration of the purified IL1beta?

Line 592, please confirm 3.6mg/L or 3.6 mg/ml.

Line 597, what was the ratio of 1:60? Mass or molar ?

Line 679, what was the cryo condition?

Line 685, what template was used in phaser?

What kind of difference maps were calculated and used for modeling the compound?

Reviewer #3 (Remarks to the Author):

The work by Hommel et al. identifies the first low molecular weight antagonist which disrupts IL-1R1 signaling, with important implications for the development of hIL-1beta-directed therapies. The authors optimize a fragment-based screening hit to improve hIL-1beta binding affinity and perform an array of biochemical and cellular assays to characterize its function. Structural analyses revealed that the binding site is found in a previously unknown cryptic pocket of mature hIL-1beta. These findings are extremely relevant for the discovery of future novel hIL-1beta-directed therapeutics and will be of benefit to ongoing efforts which aim to target this cytokine-receptor interaction. Despite these results, this work does not adequately characterize the underlying mechanism for ligand binding. Additionally, the optimized version of the ligand has only a low micromolar binding affinity for hIL-1beta, raising concerns about its efficacy in a therapeutic setting.

Major comments:

1. The conclusion that the antagonist binding stabilizes an excited-state conformation of the hIL-1beta ensemble is not adequately supported by the CEST NMR data. The text (especially abstract, discussion, and figure captions) implies that the data show a conformational selection mechanism for ligand binding which involves stabilization of the minor conformational state. However, the raw CEST profiles do not support this interpretation, and they certainly don't exclude the possibility that binding of the ligand induces a conformational change which is entirely independent of the minor state. The authors should consider removing the sections of the paper which attempt to draw a mechanistic connection between the minor state and ligand binding or, alternatively, provide further experimental evidence indicating a causative relationship. Such evidence might involve developing a mutation in hIL-1beta which either abolishes or stabilizes the minor conformational state, then verifying that this does in fact impact ligand binding (e.g., Xie et al., Science 2020). In the absence of this experimental validation, the current data may support that the D and G loops are undergoing conformational exchange in the apo protein, and that these dynamics are impacted by the binding of the ligand. Insights from computational MD simulations or other structural modeling tools (such as Rosetta) could provide further support of an

excited state-based mechanism, as shown in previous studies (e.g., Bouvignies et al., *Science*, 2011; Stiller et al., *Nature* 2022).

2. Similar studies in other cytokine-receptor systems (e.g., Thanos et al., *JACS* 2003, *PNAS*, 2006) highlight the importance of a higher-affinity (nM-range KD) ligand for the development of a therapeutic, which is the major goal of this work. The lower affinity of the ligand in this study would likely necessitate higher dosing, raising concerns about efficacy and off-target effects in vivo. Is it possible to further optimize this ligand for a greater likelihood of pharmacological success?

Minor comments:

1. The use of SPR is mentioned in the main text and in Table 1. The authors should present the raw SPR data, including the fitted curves which were used to derive kinetic rates.

2. Assignments for hIL-1beta in any bound states (compound 1 and (S)-2) which were assigned in this study should be deposited into the Biological Magnetic Resonance Bank, with the entry number listed in the methods section.

3. The NMR spectra for the hIL-1beta-(S)-2 complex and binding site mapping should be reported in the main text since this complex is the focus of all other structural analyses in the paper, whereas the hIL-1beta-compound 1 complex spectra and binding site mapping would be better suited for the supplemental data.

4. The ligand optimization procedure outlined in the text and Table 1 is confusing. It would be helpful if the authors could add more detail to the table (or show a schematic) outlining which moieties were targeted and why. Is it the case that each of the compounds 3-8 are derivatives of compound 1, and the relevant moieties from each of those were combined to generate (S)-2?

Point-by-point Response

Reviewer #1:

- What are the noteworthy results?

Hommel et al. report the discovery of the first low molecular weight antagonist of human interleukin-1 β . Despite the existence of protein antagonists of IL-1 β signaling that are currently in clinical use (e.g., Anakinra), no small molecule antagonist (with its potential for oral administration) has been reported to date. The authors characterized binding of the antagonist to the cytokine, discuss possible molecular basis for the inhibition and assess its activity through cell-based assays.

- Will the work be of significance to the field and related fields? How does it compare to the established literature?

Probably not. While the authors state that it is the first report of a low molecular weight antagonist of the IL-1 β signaling. However, several mitigating factors exist:

(1) most importantly, the novelty of these findings is substantially reduced by a publication from a few years ago (Reference 27 in this manuscript) reported a small molecule inhibitor of IL-36 γ signaling, which binds in a very similar site on the cytokine as does these authors' small molecule on IL-1 β – the larger IL-1 family of cytokines includes the IL-1 and IL-36 cytokine sub-families, which make similar signaling complexes;

The IL-1 β mode-of-inhibition of our antagonist is unique and could not be predicted by:

- 1) Inspection of apo IL-1 β
- 2) A published fragment-based screen (ref 16)
- 3) Inspection of any other IL-1 family member, including the IL-36 γ /antagonist complex (ref 32)

This is demonstrated by the high species and isoform selectivity seen with the IL-36 and IL-1 β antagonists. We have modified the relevant Supplementary Figure showing the relative locations of the IL1 β and IL-36 γ antagonists and added a sequence alignment. The absence of activity of our compound on IL-36R signaling further speaks against a very similar site in IL-1 β and IL-36 γ .

(2) a number of protein-based antagonists already exist and are in the clinical use;

The clinical use of protein-based antagonists is limited to subcutaneous/intravenous administration. The full potential of anti hIL-1 β -directed therapies may therefore be exploited only with low-molecular weight therapeutics that can penetrate e.g., brain or deep tissue (see also comment #3).

(3) the modest IC₅₀ (low micromolar) of the compound and lack of in vivo efficacy/toxicity data suggest a long road to the development of a compound that be of significance to the field or related fields.

We acknowledge that the antagonist described in our manuscript is not a clinical compound due to its moderate affinity for the target. Rather, we consider the antagonist to be a chemical probe that shows the tractability of IL-1 β as a low-molecular weight drug target. We believe this is an important discovery as biologicals and antibodies have shown limited access to tumoral tissue and do not easily pass the blood-brain barrier [Leveque, D. Wisniewski, S. and Jehl, F.; *Pharmacokinetics of Therapeutic Monoclonal Antibodies Used in Oncology; Anticancer Research* **25**, 2327-2344 (2005). Lampson, L.A. *Monoclonal antibodies in neuro-oncology. mAbs* **3**, 153–160 (2011)]. Our results will prompt future drug discovery efforts aiming at overcoming the limitations of current antibody and biological approaches.

- Does the work support the conclusions and claims, or is additional evidence needed?

No. Concerns, both major and minor, listed below:

Line 46: “conformationally excited” – do the authors mean conformationally dynamic? Flexible? Excited suggests that there is some force acting upon this part of the molecule.

There is no ‘force’ acting on the minor form. Both the major and the minor forms are part of an equilibrium. Loops in proteins can appear disordered in x-ray structures for a variety of reasons (e.g., intrinsic flexibility, lattice disorder). This kind of disorder is to be distinguished from the situation where two conformationally well-defined states, which interconvert slowly (millisecond time scale), are adopted. The study of such ‘conformationally excited states’, which are often present in only small amounts, has been pioneered by the work of Kay et al (ref 28, 30). Today we know from the work by Xie et al (ref 42), Knoverek et al. (ref 40), and Pegram et al. (ref 41) that such states represent druggable binding sites, which are not discernable by inspection of their respective apo x-ray structures. Our work thus adds an example to this growing field and shows that the region involved is also part of the binding site of our IL-1 β antagonist.

Line 119: Conventionally, the loops in IL-1 family cytokines are not labeled with letters, but with the following nomenclature: e.g., β 4-5 loop (i.e., the loop between β strands 4 and 5).

This has been adapted in the revised manuscript.

Figure 2B: The orientation of the IL-1 β /IL-1RI complex shown is strange and has no

bearing to the membrane. It is conventionally shown with the D3 domain of IL-1RI at the bottom (as if coming out of a membrane below it).

This has been adapted in the revised manuscript.

Lines 134-140: Is there really no data shown for all of these binding experiments? There appears to only be a summary of the data in Table 1. It is therefore impossible to judge whether the data is valid.

We have now added data showing ^{13}C -NMR (protein binding), ^{19}F -NMR (competition assay) and SPR in the supplementary information file.

Table 1: not immediately clear what parameter is shown.

All columns are clearly marked with what is shown.

Lines 144-145: "The detailed description of the structure-based design effort and SAR will be presented elsewhere". Is this a joke? Are we supposed to just skip over the middle half of the paper and perhaps one day (or never) read about it elsewhere?

We have added data for more compounds that we found important in our optimization process. In addition, we now provide an additional figure that should help follow the optimization strategy.

Fig. 3 B and C: The authors test their inhibitor biological activity in the cell-based assays in which they compare it to the existing biologics. What is the rationale for selection of the concentrations of all the antagonist for the assay? It would make more sense to test the same range of concentrations of particular reagents against a constant concentration of the cytokines. This would allow them to establish and compare IC₅₀ values of all the tested agents.

Biologics that antagonize IL-1 β signaling, such as canakinumab and anakinra, are well characterized. In Figure 3 (panel C and D), we used them at the single concentration of 3 nM in the presence of increasing concentrations of IL-1 β to benchmark compound (S)-2. To address this reviewer's comment, we have now provided an additional panel (panel B), which allows for direct comparison of the potency and efficacy of compound (S)-2 vs. canakinumab. The experiments that established the IC₅₀ of compound (S)-2 are recapitulated in Table 1.

In Figure 3, a legend identifying the data points would be helpful.

We have now added a legend directly into the figure.

Lines 213-225: The authors list all of the interactions between their small molecule

and IL-1 β that they observe in their crystal structure of the complex. They ascribe weight to some (“Key hydrogen bonds...”). However, they have no data indicating which intermolecular interactions are energetically important. To do so, they would need to either make a series of site-directed mutants in IL-1 β or derivatives of their small molecule in which individual functional groups were removed, and then measure binding affinities. Without such an analysis, it cannot be said that they have defined the molecular basis of this intermolecular interaction.

More SAR data highlights the relative contributions of the interactions described.

Figure 5 title: “excited” is not an appropriate descriptor here. The authors data indicate that IL-1 β exists in an ensemble of conformations, of which there are predominantly two – a “major” and a “minor” conformation, distinguished largely by the positions of residues in the loop starting at Val47. This region of IL-1 β may be conformationally flexible or dynamic, but it is not excited.

In the context of conformational equilibria, the term ‘excited’ is used in literature to describe the presence of two states, where one has a low abundance (minor form). For hIL-1 β , the equilibrium constant between the major and the minor form is $K = 11.3$ at 309 K. The minor form is thus energetically disfavored by 1.49 kcal/mol according to the equation $\Delta G(\text{major-minor}) = RT \ln(K)$.

Line: 505: from previous work it seems that number of residues involved in site A and site B is similar

The text has been changed.

- Are there any flaws in the data analysis, interpretation and conclusions? - Do these prohibit publication or require revision?

See remarks above.

- Is the methodology sound? Does the work meet the expected standards in your field?

See remarks above.

- Is there enough detail provided in the methods for the work to be reproduced?

Yes.

Reviewer #2:

In this manuscript, Hommel et al. reported the structural and functional studies of a novel small molecule inhibitor against human IL1 β . Overall, the study is thorough with extensive biophysical and functional analyses. The result is very exciting

because this is the first report to date a functional specific small molecule compound was developed targeting hIL1 β . The authors showed that the inhibitor allosterically binds to the cytokine in an isoform specific manner and blocked IL1 β signaling. The study provided new insights into structure based inhibitor design against human disease related to IL1 β pathway.

1. An insightful discussion on how the cytokine discriminate the stereoisomers of the inhibitor 1, and how (s)-2 discriminate beta from alpha IL-1 would be beneficial to the audiences.

The text has been adapted to reflect these points.

- 1) Important interactions of the fragment are made by the C-ring, including notably a short H-bond between the phenol hydroxyl and the main-chain carbonyl of Met-95. Only the (*S*)-enantiomer of the fragment, but not its (*R*)-enantiomer, can form this hydrogen bond and other hydrophobic interactions between the C-ring, Val100 and Lys97
- 2) Conversely, the para-chloro-phenyl moiety of the (*R*)-enantiomer would sterically clash with loop D.
- 3) The structural basis for the discrimination between IL-1 β and IL-1 α is provided in Supplementary Fig. 8, with a sequence alignment showing the lack of conservation of the binding site in human IL-1 α

2. In general, use x in scientific notations and use comma as thousands separator.

The text has been adapted accordingly.

3. In general, use superscripts for specific sidechain atoms. i.e. Lys-55 Nz. Use main chain carbonyl oxygen for C=O of a residue and amide for HN.

The text has been adapted accordingly.

4. Either use 'side-chain' or 'sidechain', be consistent throughout.

'Side-chain' is now used throughout the text.

5. Line 254, 43 Å.

Unit has been added.

6. All kinetic and binding values are missing standard deviations, such as Ki, Kd, IC50 etc.

Standard deviations have been added.

7. SPR titration data not shown.

SPR titration data are now shown in the Supplementary information file.

8. Fig.2A, label Loops D and G with pointer or similar to make them better discernable.

Fig.2B, add loops D and G, binding site A, B that were mentioned in the main text.

The new figures include these labels

9. Fig. 4A, indicate the 11 Å swing. Use a different color scheme for (s)-2 from the protein. Label ring C. M95 and other residues described in contacts are not visible in the figure.

10. Fig. 5. Label loop D

The figure has been adapted.

11. Table S1. Add wavelength. Keep only 1 digit after decimal point for cell contents. List beta angle only. Keep 1/100 Å precision for resolution. 1/1000 for R values.

All requested changes/additions in Table 1 have been implemented.

Why B201 has much lower B than A201?

The lower B-factor of B201 compared to A201 correlates with lower protein B-factors of nearby residues interacting with the ligand (loop B, D and G). The given values are correct.

List Wilson B. Define Rwork/Rfree.

Wilson B factor has been added to Table 1. References defining all crystallographic statistical indicators have been added to the table legend.

12. The Methods section seems very roughly written and needs careful proof reading, it contains numerous typos. Here only list some:

Line 563, italicize *E.coli*.

The Method section has been carefully edited and improved.

Line 564/570/597/599, °C

Symbol was added.

Line 574/584, mm

Unit spelling has been corrected.

Line 585, 682, leave space between a number and its unit, i.e. 50 mM, 5 ml, 260 mm

Done.

Line 579, what was the concentration of the purified IL1beta?

The concentration has now been added.

Line 592, please confirm 3.6mg/L or 3.6 mg/ml.

3.6 mg/ml is the correct value. The typo has been corrected.

Line 597, what was the ratio of 1:60? Mass or molar ?

It is a mass ratio and was entered in the text accordingly.

Line 679, what was the cryo condition?

As described in the corresponding method paragraph, no cryoprotectant was used: "One crystal was *directly* mounted in a cryo-loop and flash-cooled ..."

Line 685, what template was used in phaser?

An *in-house* structure of human IL-1 β was used as a template for molecular replacement, but this was not critical (for instance PDB entry 2I1B would work as well)

What kind of difference maps were calculated and used for modeling the compound?

The standard σ_A -weighted 2mFo-DFc and mFo-DFc maps computed by autoBUSTER were used for model building and refinement. This detail has been added to the Method paragraph.

Reviewer #3:

The work by Hommel et al. identifies the first low molecular weight antagonist which disrupts IL-1R1 signaling, with important implications for the development of hIL-1beta-directed therapies. The authors optimize a fragment-based screening hit to improve hIL-1beta binding affinity and perform an array of biochemical and cellular assays to characterize its function. Structural analyses revealed that the binding site

is found in a previously unknown cryptic pocket of mature hIL-1 β . These findings are extremely relevant for the discovery of future novel hIL-1 β -directed therapeutics and will be of benefit to ongoing efforts which aim to target this cytokine-receptor interaction. Despite these results, this work does not adequately characterize the underlying mechanism for ligand binding. Additionally, the optimized version of the ligand has only a low micromolar binding affinity for hIL-1 β , raising concerns about its efficacy in a therapeutic setting.

Major comments:

1. The conclusion that the antagonist binding stabilizes an excited-state conformation of the hIL-1 β ensemble is not adequately supported by the CEST NMR data. The text (especially abstract, discussion, and figure captions) implies that the data show a conformational selection mechanism for ligand binding which involves stabilization of the minor conformational state. However, the raw CEST profiles do not support this interpretation, and they certainly don't exclude the possibility that binding of the ligand induces a conformational change which is entirely independent of the minor state. The authors should consider removing the sections of the paper which attempt to draw a mechanistic connection between the minor state and ligand binding or, alternatively, provide further experimental evidence indicating a causative relationship. Such evidence might involve developing a mutation in hIL-1 β which either abolishes or stabilizes the minor conformational state, then verifying that this does in fact impact ligand binding (e.g., Xie et al., Science 2020). In the absence of this experimental validation, the current data may support that the D and G loops are undergoing conformational exchange in the apo protein, and that these dynamics are impacted by the binding of the ligand. Insights from computational MD simulations or other structural modeling tools (such as Rosetta) could provide further support of an excited state-based mechanism, as shown in previous studies (e.g., Bouvignies et al., Science, 2011; Stiller et al., Nature 2022).

We have generated and characterized the variant hIL-1 β^{V47A} both in terms of its ligand binding kinetics and its conformational properties. We can show that the amount of minor form present in solution is thereby shifted ~3-fold, which is corroborated by a decreased amount of protein required in the ^{19}F -transverse relaxation experiments. Furthermore, the variant shows a ~3-fold increased on-rate for the parent fragment, suggesting a facilitated access to the cryptic pocket. Together, these data provide a mechanistic link between the minor form and binding of ligands to the cryptic pocket.

2. Similar studies in other cytokine-receptor systems (e.g., Thanos et al., JACS 2003, PNAS, 2006) highlight the importance of a higher-affinity (nM-range KD) ligand for the development of a therapeutic, which is the major goal of this work. The lower affinity of the ligand in this study would likely necessitate higher dosing, raising

concerns about efficacy and off-target effects in vivo. Is it possible to further optimize this ligand for a greater likelihood of pharmacological success?

We acknowledge that the antagonist described in our manuscript is not a clinical compound due to its moderate affinity for the target. Rather, we consider the antagonist to be a chemical probe that shows the tractability of IL-1 β as a low-molecular weight drug target. We believe this to be an important discovery as biologicals and antibodies have shown limited access to tumoral tissue and do not easily pass the blood-brain barrier [Leveque, D. Wisniewski, S. and Jehl, F.; Pharmacokinetics of Therapeutic Monoclonal Antibodies Used in Oncology; Anticancer Research **25**, 2327-2344 (2005). Lampson, L.A. Monoclonal antibodies in neuro-oncology. mAbs **3**, 153–160 (2011)]. Our results will prompt future drug discovery efforts aiming at overcoming the limitations of current antibody and biological approaches.

In addition to selectivity against IL-1 α (Fig. 3), we are now providing selectivity data against IL-36 γ (Supplementary Fig. 9), which further demonstrates the selectivity of our compound/mechanism despite its limited potency.

Optimization of the antagonist would profit from finding alternative and/or synergistic binders that may arise from screening DNA-encoded libraries, click chemistry approaches, virtual ligand screening. The knowledge of the structure presented in our manuscript will thus be a guide to others in any rational approach targeting the cryptic pocket.

Minor comments:

1. The use of SPR is mentioned in the main text and in Table 1. The authors should present the raw SPR data, including the fitted curves which were used to derive kinetic rates.

SPR data have been added as Supplementary Information.

2. Assignments for hIL-1beta in any bound states (compound 1 and (S)-2) which were assigned in this study should be deposited into the Biological Magnetic Resonance Bank, with the entry number listed in the methods section.

The access codes for the respective BMRB entries are provided.

3. The NMR spectra for the hIL-1beta-(S)-2 complex and binding site mapping should be reported in the main text since this complex is the focus of all other structural analyses in the paper, whereas the hIL-1beta-compound 1 complex spectra and binding site mapping would be better suited for the supplemental data.

We suggest keeping the hIL-1 β /compound 1 complex spectra and binding site mapping in the main section as they show an important aspect of our work for the first time, i.e., the link between a slow binding process and the location of the binding site involved.

Furthermore, we have added the hIL-1 β /(S)-2 complex spectra plus binding site mapping as a Supplementary Figure and refer to this in the section where we describe the structural details of (S)-2 binding.

4. The ligand optimization procedure outlined in the text and Table 1 is confusing. It would be helpful if the authors could add more detail to the table (or show a schematic) outlining which moieties were targeted and why. Is it the case that each of the compounds 3-8 are derivatives of compound 1, and the relevant moieties from each of those were combined to generate (S)-2?

More details and a new chemical optimization scheme have been added. Indeed, all compounds are derivatives or analogues of compound 1.

REVIEWERS' COMMENTS

Reviewer #1 (Remarks to the Author):

The authors have made reasonable attempts to address the reviewers' critiques.

Reviewer #2 (Remarks to the Author):

The revised manuscript has addressed the concerns from this reviewer.

Reviewer #3 (Remarks to the Author):

The authors have provided additional new results and analysis which further bolsters their conclusions regarding the role of a minor state, as identified by NMR. In addition, they have performed significant revisions of the text, and provided the missing data in the supplement. Together, these revisions address my concerns, therefore I recommend publication of the work in Nature Communications and congratulate the authors for their contribution.